# Full-Length RNA Sequencing Provides Insights into Goldfish Evolution under Artificial Selection

**DOI:** 10.3390/ijms24032735

**Published:** 2023-02-01

**Authors:** Xuedi Du, Weiwei Zhang, Jiali Wu, Congyuan You, Xiaojing Dong

**Affiliations:** College of Animal Science and Technology, Yangzhou University, Yangzhou 225009, China

**Keywords:** goldfish, full-length transcriptome, alternative splicing, alternative polyadenylation, gene fusion, differentially expressed genes

## Abstract

Goldfish *Carassius auratus* is an ideal model for exploring fish morphology evolution. Although genes underlying several ornamental traits have been identified, little is known about the effects of artificial selection on embryo gene expression. In the present study, hybrid transcriptome sequencing was conducted to reveal gene expression profiles of Celestial-Eye (CE) and Ryukin (RK) goldfish embryos. Full-length transcriptome sequencing on the PacBio platform identified 54,218 and 54,106 transcript isoforms in CE and RK goldfish, respectively. Of particular note was that thousands of alternative splicing (AS) and alternative polyadenylation (APA) events were identified in both goldfish breeds, and most of them were inter-breed specific. RT-PCR and Sanger sequencing showed that most of the predicted AS and APA were correct. Moreover, abundant long non-coding RNA and fusion genes were detected, and again most of them were inter-breed specific. Through RNA-seq, we detected thousands of differentially expressed genes (DEGs) in each embryonic stage between the two goldfish breeds. KEGG enrichment analysis on DEGs showed extensive differences between CE and RK goldfish in gene expression. Taken together, our results demonstrated that artificial selection has led to far-reaching influences on goldfish gene expression, which probably laid the genetic basis for hundreds of goldfish variations.

## 1. Introduction

Originating from the crucian carp in the lower reaches of the Yangtze River [1,2], and extended to the whole world in the past five centuries for ornamental purposes, the goldfish is world famous for its morphological attractiveness. To biologists, however, even more interesting and fantastic is its fast evolution under artificial selection [3,4], which has resulted in hundreds of morphological variations. For detailed information about the hundreds of goldfish breeds, please refer to the book “Goldfish of China” by Ye and Qu [5].

According to the morphology, goldfish are traditionally divided into three to five categories [6], among which the three-breed taxonomy, i.e., Grass-goldfish, Wen-goldfish and Egg-goldfish, better indicates the history of domestication [2]. The Grass-goldfish is very similar to native *Carassius auratus*, except in the color of its scales, while Wen- and Egg-goldfish own many different variations in their head, tail-fin, eye, scale, and so forth. The condition of the dorsal fin (lost or retained) distinguishes the Egg-breed from the Wen-breed [6]. According to Chinese texts that date to the Song dynasty, goldfish breeding for ornamental purposes can be dated back to one thousand years ago, which makes it the most historic fish species to undergo artificial selection. In the past 500 years since the Ming dynasty, a large number of morphologically diverged breeds have formed, suggesting a huge effect of artificial selection on goldfish morphology evolution. The abundant variations combined with a clear history of artificial selection make goldfish an ideal model for exploring fish morphology evolution [4]. To date, the developmental process of goldfish has been well recorded [3,7], and the complete genome has been sequenced by several institutions around the world [8,9], which has paved the way for dissecting the genetic basis underpinning goldfish morphology evolution.

Recently, it has been determined that the bifurcated caudal skeleton arose from a stop-codon mutation in the *chordinA* gene, which affects embryonic dorsal–ventral (DV) patterning [10]. Although depletion of both *szlA* and *szlB* genes in single-fin goldfish was also able to produce the bifurcated caudal fin found in twin-tail goldfish, szl-type twin-tail goldfish has not been observed in domesticated goldfish populations [11]. In medaka fish, the deletion of a zone of polarizing (ZPA) regulatory sequence targeting *shh* gene causes an almost complete ablation of the dorsal fin [12], even though the median fin fold was not affected. Our observation on the goldfish embryo development process revealed, however, that the dorsal fin fold does not form at all in Egg-goldfish, which is first morphologically recognizable at the late somite stage under a stereomicroscope. This big difference let us suppose that dorsal fin loss in Egg-goldfish has a different developmental mechanism compared to dorsal fin ablated medaka fish. Most recently, a genome wide association study revealed that the deletion of a 313 bp fragment in the *lrp6S* gene was probably the causative variation leading to dorsal fin loss phenotype in goldfish [13], and it was further verified in zebrafish that partial inhibition of the Wnt pathway mediated by Lrp6 affects dorsal fin formation. However, whole genome sequencing and GWAS study conducted by another team identified as many as 378 genes as candidates for dorsal fin loss phenotype in goldfish, and the *lrp6* gene was not included [9]. Given that goldfish have undergone over five hundred years of artificial selection, leading to production of hundreds of breeds with different morphology variations, the paradox between the two GWAS studies probably reflected the complex genetic basis underlying goldfish morphology evolution. We cannot refrain from asking, to what extent do different goldfish breeds differ at transcription level?

In the present study, a comparative transcriptome sequencing was conducted using Celestial-Eye (CE) and Ryukin (RK) goldfish embryos. Both whole length transcriptome sequencing and RNA-seq were used. Our study identified abundant inter-breed specific transcript isoforms, and highlighted the role of alternative splicing, and polyadenylation and gene fusion in producing inter-breed specific transcripts.

## 2. Results

### 2.1. Whole Length Transcript Profile of Goldfish Embryos

Total RNA from three stages of embryos, i.e., zygote, 14-somite, and 35% OVC, were equally mixed and subjected to library preparation for CE and RK, respectively. Here, we report that the whole length transcriptome sequencing on the PacBio sequel platform yielded 4,517,802 and 4,947,400 reads in CE and RK, respectively, and of which 260,355 and 272,307 were high quality circular consensus sequences (CCS). About 80% of the CCS were full-length non-chimeric reads (Flnc). Then, Flnc reads were further clustered, leading to the identification of 117,201 and 122,576 consensus reads in CE and RK, respectively, which were again corrected using the next generation sequencing data. Finally, we identified 117,201 and 122,576 polished consensus reads with average length of 3149 and 3147 base pairs in CE and RK, respectively (Table 1).

The polished consensus reads were compared against the goldfish reference genome RefGen_v2 using GMAP software (version 2017-11-15, Genentech, South San Francisco, CA 94080, USA). More than 95% of the reads were mapped to the reference, of which over 76% were uniquely mapped. All the mapped reads showed more than 98% identity with the reference, while more than 81% of the total reads showed high coverage in Ryukin goldfish (Figure 1a,b). Polished consensus reads mapped to each chromosome were also counted. All reads were quite evenly distributed on the genome (Figure 1c), and about 60% and 55% of the uniquely mapped consensus transcripts were from the plus strand in CE and RK, respectively.

Polished consensus sequences were further analyzed using TAPIS pipeline, leading to the identification of 54,218 and 54,106 isoforms in CE and RK goldfish, respectively. Combining the two sets of data, we finally obtained a total of 93,308 isoforms from 44,211 unique genes (Table 2).

Of these genes, more than one third produced at least two transcripts in both CE and RK embryos. Of particular note was that less than 10 percent of the identified transcripts were reported previously, while about 70 percent of the transcripts were novel transcripts from known genes. A total of 41,830 and 38,273 isoforms were identified as multi-exon in CE and RK goldfish, respectively, which were likely to be true full-length transcripts. Compared with the reference genome, PacBio Iso-seq recovered more exons (Figure 2a), and the average length of identified isoforms was much longer (Figure 2b). In addition, over 20 percent of the identified transcripts were from novel genes. Of the identified transcripts, about 95% accorded with the GU-AG rule, while 0.81% transcripts were spliced at the GC-AG site. The unmapped reads and the predicted novel genes were annotated by BLAST analysis against seven key databases (NR, NT, Pfam, KOG/COG, SwissProt, KEGG and GO) (Appendix A).

### 2.2. Alternative Splicing Analysis

We detected 29,829 AS events produced from 11,426 genes, accounting for 25.8% of the total sequenced genes (44,211). Of the identified AS events, 2754 were CE specific, while 2960 were RK specific (Appendix A). The most abundant AS types were skipping exon (SE), alternative 5’ splice-site (A5), alternative 3’ splice-site (A3), retained intron (RI), and alternative first exon (AF), each accounting for 16% to 23% of the total AS events (Appendix A). A total of 8224 AS events were detected in both CE and RK, accounting for 27.6% of the total events. The proportion of each type of common AS event varied, ranging from less than 10% for RI to over 40% for SE and maturely exclusive exons (MX).

Five genes were randomly selected to validate the predicted AS events by RT-PCR, and amplicons were verified by Sanger sequencing. Here, we show that four out of these five genes produced eleven different transcripts in three embryonic stages of common, CE and RK goldfish (Figure 3), and that some transcripts showed breed or stage specific expression. For example, the longest transcript of CA000831 that harbored all exons was not detectable in common goldfish 30%-epiboly and RK 14-somite stages. On the other hand, none of the four alternatively spliced transcripts from CA002148 and CA001126 were detectable in RK and CE 14-somitethe embryos. Moreover, the abundance of alternatively spliced isoforms varied.

### 2.3. Alternative Polyadenylation Analysis

In the present study, 14,756 and 14,116 transcripts with poly A structure were identified in Ryukin and Celestial-Eye goldfish embryos, respectively. These detected transcripts were mapped to 13,337 genes in total, among which 7668 genes generated poly A mRNA in both of the two breeds. Similar to AS, alternative polyadenylation (APA) leads to the production of more than one transcript from a single gene. For example, hundreds of genes produced more than three transcripts by polyadenylation at different sites. A total of 2639 and 3016 genes yielded 6214 and 7308 poly A mRNA in Celestial-Eye and Ryukin goldfish embryos, respectively (Appendix A). In addition, as many as 2605 genes showed different patterns between the two breeds (Appendix A).

To validate the predicted APA events during goldfish embryogenesis, several genes showing high level expression were randomly selected, and RT-PCR was conducted using 3’ rapid amplification of cDNA ends (3’RACE). According to gel electrophoresis and Sanger sequencing, all predicted APA events were verified (Figure 4). In addition, different APA patterns were observed in different breeds and different developmental stages.

### 2.4. Long Non-Coding RNAs

We identified 3325 and 5203 long non-coding RNA (lncRNA) in Ryukin and Celestial-Eye goldfish, respectively (Figure 5). About two thirds of them were transcribed from inter genic regions, and almost 60 percent of lncRNA in gene regions were located on the negative chain. It was worth noting that only 265 lncRNA were shared between the two breeds. In other words, over 90% of the identified lncRNA were different between the two breeds.

### 2.5. Gene Fusions

We identified 912 and 847 gene fusion events in Ryukin and Celestial-Eye goldfish, respectively (Figure 6, Appendix A). About half of the fusion transcripts were formed by blocks located on assembled chromosomes. Most of these fusions were inter-chromosome type, and the percentages were 375 out of 493 and 317 out of 431 in Ryukin and Celestial-Eye, respectively. Among these fusion genes, 48 were shared by the two breeds (Appendix A).

### 2.6. Differentially Expressed Genes between Celestial-Eye and Ryukin

RNA-seq on the Hiseq 2000 platform produced over 40 million reads for each replicate of the three stages, i.e., zygote, 14-somite, and 35%-OVC, in both Celestial-eye and Ryukin, and more than 93% of the total reads were mapped to the goldfish genome on average (Appendix A). Pearson correlation analysis showed that gene expression levels between replicates of each stage were in good consistence in both Celestial-eye and Ryukin (R^2^ > 0.92, Appendix A). When compared between different stages, however, the coefficient square value varied, ranging from 0.3 to 0.89. In particular, the R^2^ value between the zygote and other stages was much lower (0.3 to 0.5). The analysis on genes’ FPKM value showed that more than 50% of genes were transcribed at a low level (FPKM < 0.1), especially in the zygote stage (Appendix A). On the other hand, however, the number of highly expressed genes (FPKM > 60) in the zygote stage was much larger than the other two stages, and decreased along with the embryo’s development.

Genes with an FPKM value less than 0.1 were treated as having non-effective expression, and were not included in differential expression analysis between different breeds and developmental stages. We show that many genes showed a developmental stage specific expression pattern (Appendix A). For example, over fifteen thousand genes were up-regulated while over ten thousand genes were down-regulated in fourteen somite embryos compared with zygotes in both Celestial-Eye and Ryukin. More importantly, we identified thousands of genes that showed breed specific expression at each embryonic stage (Table 3). At the 35%-OVC stage, for example, 7101 and 7250 genes were specifically expressed in Celestial-Eye and Ryukin, respectively. As for genes that showed expression in both Celestial-Eye and Ryukin, about 12 percent were either down-regulated or up-regulated in Celestial-Eye goldfish zygote and 14-somite stages (Figure 7). In the 35%-OVC stage, however, only about six percent of detected genes were differentially expressed.

Then, differentially expressed genes between the two breeds at the three stages were combined, and further subjected to cluster analysis (Figure 8). Obviously, DEGs in the three stages were categorized into five clusters, among which cluster A and cluster B showed high expression levels mainly in goldfish zygotes, while clusters C, D, E were mainly expressed in 14-somite and 35% OVC stages. On the other hand, the gene expression pattern of RK 14-somite was clustered with RK 35% OVC into a clade, while CE 14-somite was clustered with CE 35% OVC into another clade. Then, the two clades were further clustered. Meanwhile, the CE zygote showed a similar expression pattern to the RK zygote.

### 2.7. DEGs Enrichment Analysis

Overall, we detected 6259, 8862, and 4805 differentially expressed genes (DEGs) in zygote, 14-somite, and 35%-OVC embryos, respectively, between the two breeds. All genes with KEGG annotations were subjected to KEGG pathway enrichment analysis using omicshare tools (https://www.omicshare.com/tools/Home/Soft/pathwaygseasenior (accessed on 17 November 2022)). As shown in Figure 9, the gene expression profile between the two goldfish breeds differed in many ways. At the zygote stage, most DEGs are involved in genetic information processing pathways. With embryo development, however, the majority of DEGs are involved in metabolism pathways.

## 3. Discussion

Hundreds of years of artificial selection have led to hundreds of different goldfish breeds. Although the goldfish genome project has been completed for years [8] and several genes underlying some ornamental traits have been identified by whole genome association studies [9,13], little is known about the effects of artificial selection on gene expression patterns, especially at embryonic stages. Given that most ornamental traits might have been determined in the embryonic development stages, comparative transcriptome sequencing of embryos at different developmental stages seemed to be an appealing approach to reveal the effect of artificial selection on goldfish evolution at the transcription level. In the present study, a hybrid strategy combining single-molecular real-time (SMRT) sequencing and next generation RNA-seq is adopted to recover whole length transcripts and conduct quantitative analysis at the same time [14,15,16].

Here, we report that PacBio iso-seq produced over 93 thousand transcript isoforms from 44,211 genes in Celestial-Eye and Ryukin goldfish, and more than one third of the detected genes produced at least two transcript isoforms in both goldfish breeds. In addition, as many as 70 percent of the identified transcripts were novel from known genes, as compared with the reference genome, and PacBio iso-seq recovered more exons and a longer transcript length than the reference. These findings were comparable with previous research in maize [17], sugarcane [18], and pigs [19], and underscored the advantage of PacBio sequencing in identifying full-length splice isoforms and transcript diversity. It is worth noting that most of the identified transcripts were breed specific, which was probably caused by limited sequencing depth. To some extent, however, it also reflected the wideness of differentially expressed genes between the two breeds.

Alternative splicing (AS) and alternative polyadenylation (APA), which bring in variations in gene coding regions and 3’ UTR regions, respectively, are critical means for the cell to expand protein products and regulation diversity [20,21]. In the present study, abundant AS and APA events were identified in the embryos of Celestial-Eye and Ryukin goldfish. RT-PCR analysis revealed that most of the identified post transcription modifications were correct. Of particular note is that many AS and APA were breed and embryonic stage specific. Given that transposons have been approved to be an important factor leading to new genes and new expression patterns [22], and active transposons have been identified in the goldfish genome [23], it is speculated that transposons probably contributed to the production of breed specific AS and APA. Since many studies have demonstrated that AS and APA are important mechanisms involved in development and respond to environmental stimuli [24,25,26], variations in AS and APA are believed to place a far-reaching impact on goldfish embryo development. We speculate that the identified Celestial-Eye and Ryukin specific AS and APA are potential candidates underlying some goldfish morphological variations.

It is now known that the eukaryote genome is transcribed more pervasively than previously thought, but a very little part of the transcripts has the ability to encode proteins. Once considered as transcriptional noise, significant numbers of long non-coding RNAs (lncRNAs) have been identified as modulators of gene expression, and participate in crucial processes such as development and disorders [27,28,29]. It was worth noting that PacBio iso-seq recovered abundant lncRNAs in both Celestial-Eye and Ryukin goldfish embryos, but only 265 lncRNA were shared between the two breeds, accounting for less than 10 percent of the identified lncRNA in each goldfish breed. Since numerous lncRNAs have been determined as biomarkers of metabolic and cellular processes in humans and livestock [30,31,32], such a low ratio of shared lncRNA between the two breeds make us speculate that differentially expressed lncRNAs might participate in the regulation of goldfish morphogenesis.

Moreover, whole length transcriptome sequencing identified many gene fusion events in the two goldfish breeds, and the majority of fusions were breed specific. A gene fusion is a hybrid gene formed from two previously independent genes, occurring as a result of translocation, interstitial deletion, or chromosomal inversion, and is a major cause of cancer [33]. Although RNA-seq can be used to identify gene fusion, it has low sensitivity and high sequencing depth requirements [34,35,36]. By contrast, whole length transcriptome sequencing is more sensitive and efficient, and thus it has been widely used [37,38]. As far as we know, this is the first set of gene fusion resources in goldfish. Further research on it may give us insights into goldfish evolution.

Although whole length transcriptome sequencing showed great advantages in identifying full-length transcripts and structure variations, the low coverage and sequencing depth make it not applicable for quantitative gene expression analysis. In the present study, we also adopted RNA-seq to determine gene expression levels, and a huge difference was observed in gene expression between Celestial-Eye and Ryukin goldfish embryos. Most of the DEGs at the zygote stage were enriched in genetic information processing pathways, indicating that maternal inherited transcripts play critical roles in goldfish development.

Taken together, our result revealed a very different gene expression profile between Celestial-Eye and Ryukin goldfish embryos, which probably reflected the far-reaching influence of artificial selection on goldfish genome evolution. Further research on breed specific AS and APA events, lncRNAs, and fusion genes will deepen our understanding on goldfish morphogenesis and breed ornamental trait development.

## 4. Materials and Methods

### 4.1. Fish Samples

Two-year-old adult individuals of Ryukin (belongs to Wen-goldfish) and Celestial-Eye (belongs to Egg-goldfish) were purchased from the aquarium fish agency in Yangzhou in April 2018, and kept separately in tanks (3600 L of water in volume, dimensions of 200 × 200 cm in bottom area) in nature conditions (water temperature ranged between 15 and 22 °C). The courtship was monitored at seven to eight o’clock in the morning every day before artificial fertilization.

### 4.2. Artificial Fertilization

The naturally mature goldfish were selected for artificial fertilization. Eggs were squeezed out from one mature female into Teflon coating dishes. Immediately, sperm was taken from one male goldfish, and artificial fertilization was performed using dry methods. The fertilized eggs were spread into 9 cm Petri dishes at a low density, and subjected to incubation under a constant temperature of 24 °C until hatching. The dead individuals were removed such as to avoid water pollution.

### 4.3. Total RNA Preparation and Transcriptome Sequencing

Embryos were observed under microscopes, and developmental stages were determined according to the developmental stages of common goldfish [3,7]. Embryos at three stages, i.e., zygote period, 14-Somite stage, and 35% optic vesicle closure (OVC) stage were collected and subjected directly to RNA preparation using Trizol reagent. Five embryos were used for each stage. Total RNA quality was determined by electrophoresis. RNA from the three developmental stages were mixed equivalently, before library construction and sequencing on the PacBio platform. The Iso-Seq library was prepared according to the Isoform Sequencing protocol (Iso-Seq) using the Clontech SMARTer PCR cDNA Synthesis Kit and the BluePippin Size Selection System protocol, as described by Pacific Biosciences (PN 100-092-800-03). For gene expression analysis, RNA from each stage of Ryukin and Celestial-Eye goldfish were subjected to library preparation and sequencing separately, according to the protocol. RNA-seq was performed on the HiSeq 2000 platform.

### 4.4. Sequencing Data Processing

PacBio sequencing data were processed using the SMRTlink 5.1 software (Pacific Biosciences, Menlo Park, CA 94025, USA). Any redundancy in consensus reads was removed by CD-HIT (-c 0.95 -T 6 -G 0 aL 0.00 -aS 0.99) to obtain final transcripts for the subsequent analysis. The polished transcripts were mapped to the reference genome using GMAP software. Gene structure analysis was performed using TAPIS pipeline. The GMAP output bam format file and gff/gtf format genome annotation file was used for gene and transcript determination. Alternative splicing events and alternative polyadenylation events were then analyzed. Fusion transcripts were determined as transcripts mapping to two or more long-distance range genes.

For long non-coding RNA analysis, CNCI, CPC, Pfam-scan, and PLEK were used to predict the coding potential of transcripts. Transcripts predicted with coding potential by either/all of the four tools above were filtered out. Unmapped transcripts and novel gene transcripts were annotated based on the following databases: NR (NCBI non-redundant protein sequences), NT (NCBI non-redundant nucleotide sequences), Pfam (Protein family), KOG/COG (Clusters of Orthologous Groups of proteins), Swiss-Prot (A manually annotated and reviewed protein sequence database), KO (KEGG Ortholog database), and GO (Gene Ontology). The threshold e-value was set to 1e-10 for the above-mentioned database analysis.

Pearson correlation analysis was conducted to determine the reproducibility of gene expression between replicates. Gene expression level was determined by mapping RNA-seq reads to the reference genome. Reference genome and gene model annotation files were downloaded from the genome website directly. The index of the reference genome was built using HISAT2 (v2.1.0, https://github.com/DaehwanKimLab/hisat2, accessed on 17 November 2022), and paired-end clean reads were aligned to the reference genome. HTSeq (v0.6.1, https://htseq.readthedocs.io/en/master/, accessed on 17 November 2022) was used to count the read numbers mapped to each gene, and then the FPKM of each gene was calculated. Differential gene expression analysis was performed using the DESeq R package (1.18.0, https://www.bioconductor.org/packages//2.10/bioc/html/DESeq.html, accessed on 17 November 2022). The resulting *p*-values were adjusted using Benjamini and Hochberg’s approach for controlling the false discovery rate. Genes with an adjusted *p*-value < 0.05 were assigned as differentially expressed.

### 4.5. KEGG Enrichment Analysis

Omicshare tools (https://www.omicshare.com/tools/Home/Soft/pathwaygseasenior, accessed on 17 November 2022) were used to test the statistical enrichment of differential expression genes in KEGG pathways. KEGG pathways with a corrected *p* value less than 0.05 were considered significantly enriched. The bubble image was drawn by GraphPad Prism 9 software (GraphPad Software, San Diego, CA 92108, USA).

### 4.6. RT-PCR Analysis

Total RNA from both Celestial-Eye and Ryukin at zygote, 14-Somite, and 35% OVC stages were used for reverse transcription PCR analysis. For AS events, five genes were randomly selected, and primers were designed for the first and last exons, or appropriate regions of these genes. The gel banding pattern and the size of the fragments were compared with the detected isoforms from the Iso-Seq data. Each band was then cloned and sequenced to verify the splice junctions. For APA events, another five genes were randomly selected, and 3’ RACE was conducted. Similarly, both the gel banding pattern and fragment size were analyzed, followed by Sanger sequencing.

## Figures and Tables

**Figure 1 ijms-24-02735-f001:**
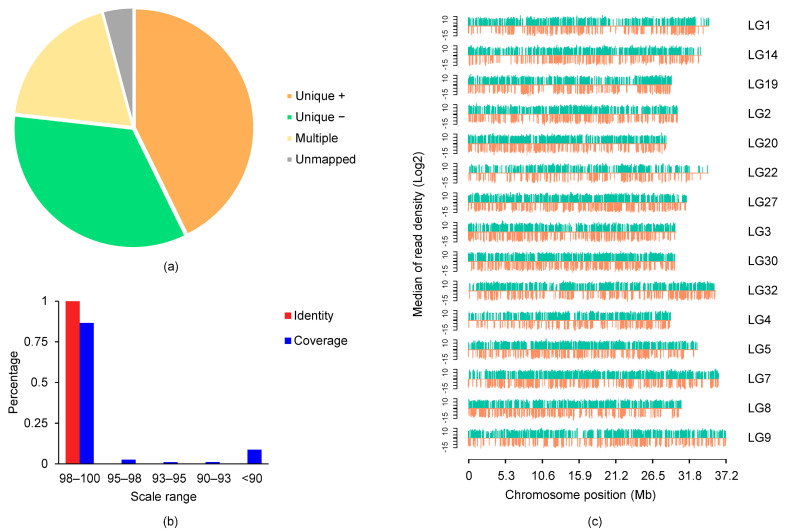
Genome wide distributions and consistencies of Ryukin consensus reads. (**a**) Mapping of consensus reads on the reference genome; (**b**) gene identity and coverage of consensus reads; (**c**) distribution of consensus reads on different chromosomes.

**Figure 2 ijms-24-02735-f002:**
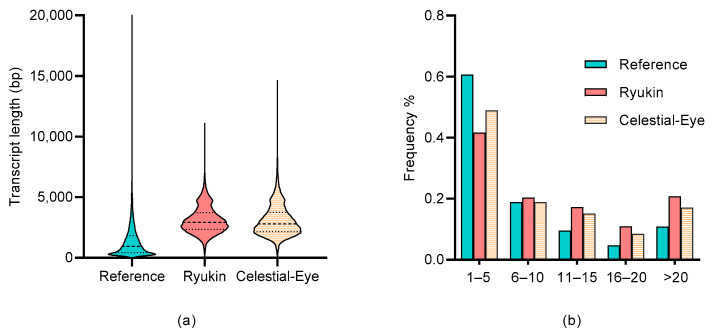
Comparison of transcript length and exon number of full-length transcriptome sequencing with reference genome. (**a**) Transcript length distribution of polished consensus reads; (**b**) the number of exons recovered by polished consensus reads.

**Figure 3 ijms-24-02735-f003:**
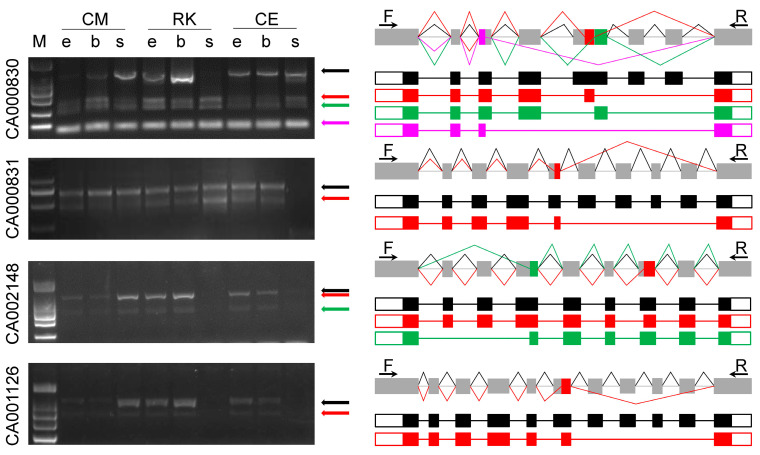
Alternative splicing verification by RT-PCR. CM, RK, and CE represent Common, Ryukin, and Celestial-Eye goldfish, respectively. e: 30% epiboly stage, b: bud stage, s: 14-somite stage. F and R represent forward and reverse primers used for RT-PCR.

**Figure 4 ijms-24-02735-f004:**
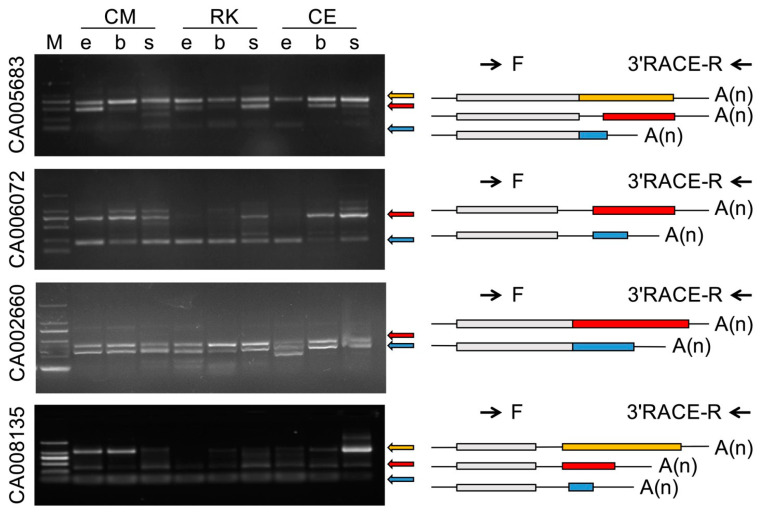
Alternative polyadenylation verification by 3’RACE. CM, RK, and CE represent Common, Ryukin, and Celestial-Eye goldfish, respectively. e: 30% epiboly stage, b: bud stage, s: 14-somite stage. F and 3’RACE-R represent forward and reverse primers used for 3’RACE.

**Figure 5 ijms-24-02735-f005:**
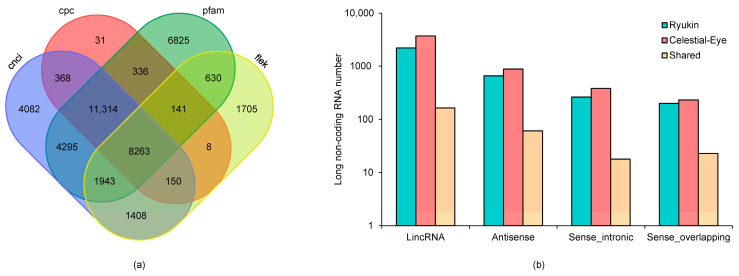
Long non-coding RNAs identified in Ryukin and Celestial-Eye goldfish. (**a**): LncRNAs predicted by different methods; (**b**) genomic distribution of lncRNA.

**Figure 6 ijms-24-02735-f006:**
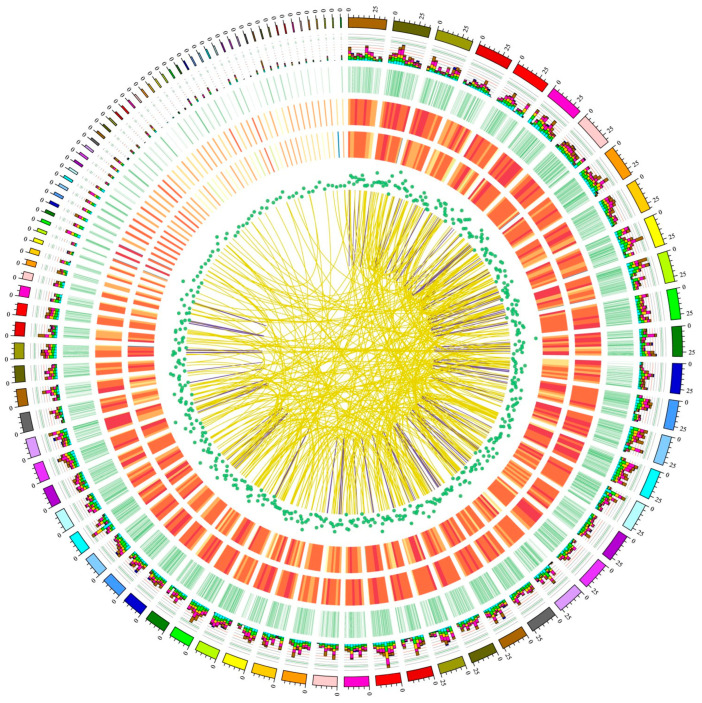
CIRCOS visualization of different data of Ryukin at the genome-wide level. From the outer to the inner are goldfish chromosomes of the reference genome, stacked histogram of alternative splicing loci (with light blue, green, yellow, purple, red, brown, and navy blue represent RI, A3, A5, SE, MX, AF, and AL, respectively), alternative polyadenylation loci, density distribution of novel transcripts identified in the present study, density distribution of novel genes identified in the present study, density distribution of lncRNAs, and gene fusions identified in Ryukin goldfish with yellow and purple lines represent intra- and inter-chromosome gene fusions, respectively.

**Figure 7 ijms-24-02735-f007:**
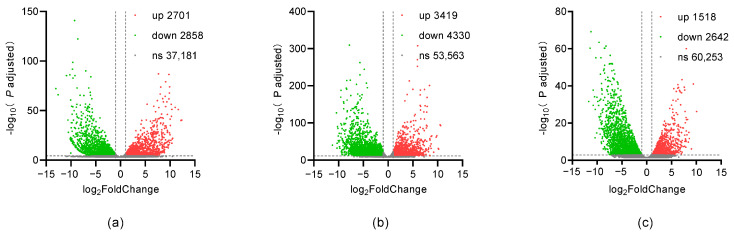
Differentially expressed genes between CE and RK goldfish at different embryonic stages. (**a**) zygote; (**b**) 14-somite; (**c**) 35% OVC.

**Figure 8 ijms-24-02735-f008:**
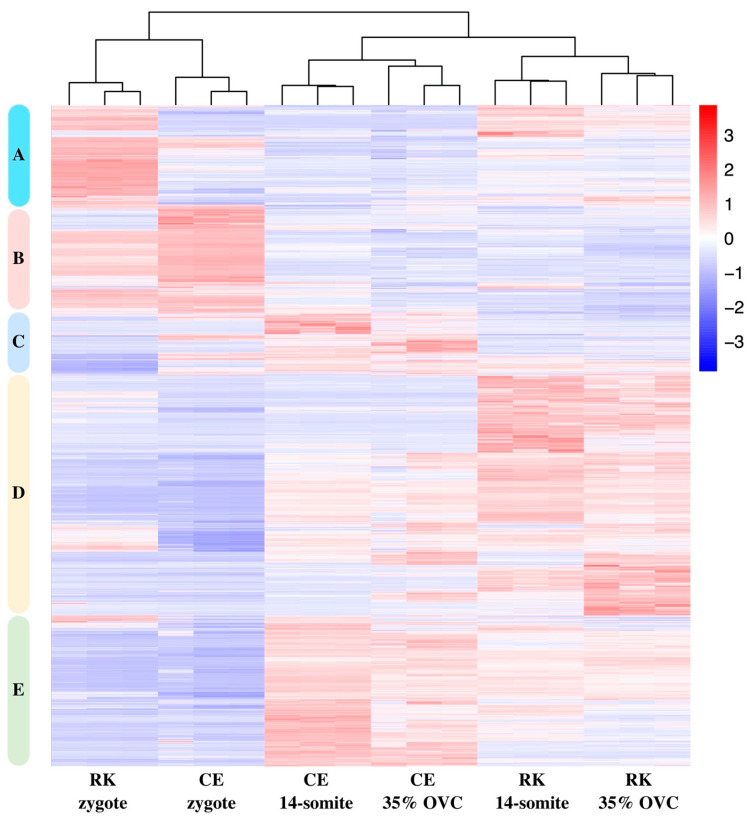
Cluster heatmap of differentially expressed genes between CE and RK goldfish. The colors correspond to gene expression levels that were converted to Z-scores.

**Figure 9 ijms-24-02735-f009:**
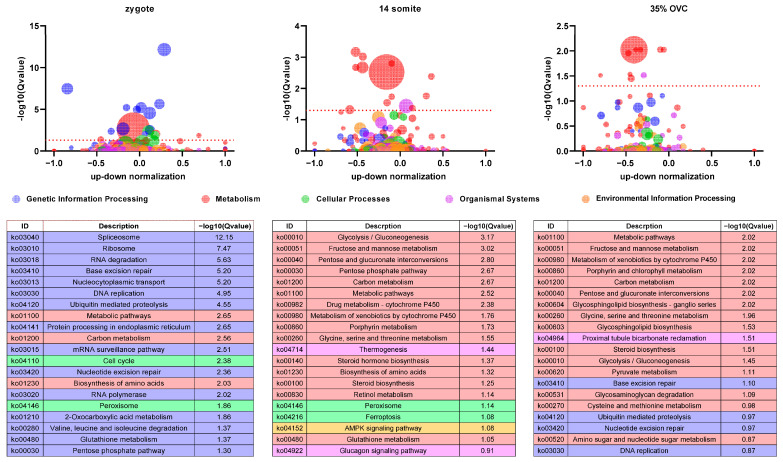
KEGG enrichment analysis of differentially expressed genes between CE and RK goldfish embryos.

**Table 1 ijms-24-02735-t001:** Summary information of full-length transcriptome sequencing.

Index	Celestial-Eye	Ryukin
CCS	260,355	272,307
Flnc	210,667	217,781
Polished consensus reads	117,201	122,576
Average consensus reads length	3149	3147
N50 consensus reads length	3311	3220

**Table 2 ijms-24-02735-t002:** Summary of identified transcript isoforms in goldfish embryos.

Index	Celestial-Eye	Ryukin	Total
Isoforms number	54,218	54,106	93,308
Isoforms of known genes	4743	4611	6088
Novel isoform of known genes	35,246	39,089	63,598
Isoforms of novel genes	14,229	10,406	23,622
Gene coverage	31,702	27,986	44,211
Isoform length N50	3325	3286	3306

**Table 3 ijms-24-02735-t003:** Breed specific gene expression at different embryonic stages.

Embryonic Stages	CE Specific	RK Specific	Shared
Zygote	5892	8961	43,126
14-somite	5586	8554	61,312
35% OVC	7101	7250	64,413

## Data Availability

Full-length transcriptome sequencing raw data can be downloaded through accession SRR9924704 and SRR9924705 for Celestial-Eye and Ryukin goldfish, respectively. RNA-seq raw data can be downloaded through accession SRR9899354-SRR9899371.

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
