# Peer review of "Full-Length RNA Sequencing Provides Insights into Goldfish Evolution under Artificial Selection"

_ijms, 2023, doi:10.3390/ijms24032735_

Round 1

Reviewer 1 Report

In this study, authors focused on the goldfish evolution under artificial selection, and tried to reveal the factor by full-length RNA sequencing. This is very interesting to understand the goldfish from the viewpoints of morphological differentiation.

Please refer comments as noted below and revise for publish.

<Discussion>

・Here, authors focused on AS and APA as candidates underlying some gold fish morphological variations. In artificial conditions, people has carried out selective breeding for a long time as authors explained in Introduction. In artificial conditions, how do authors consider  what a kind of factors did induce the gene variations including AS and APA for differentiation of Celestial-Eye and Ryukin goldfish? If possible, I suggest to add this point in Discussion.

<References>

・Please re-check the style of all reference list and unify it. For example, please check whether the name of journal was abbreviated or not, and the presence and absence of "." after abbreviated name of journal.

<Entire manuscript>

・Please re-check the style of manuscripts and unify it. For example, in some values, there are some comma "," like "8,244" in Line 126, but nothing in other cases.

Reviewer 2 Report

Abstract

This section is well prepared.

Introduction

This section is well prepared.

Results

This section is well prepared.

What samples were used for transcriptome analysis? Were all developmental stages used together? (Figures 1 and 2). Clarify this aspect.

Discussion

This section is well prepared. I would like to ask some reflection questions that should be considered in this section.

Of the genes and metabolic pathways detected, what percentage of them are related to possible genetic diseases? Genetic selection and interbreeding can generate a significant loss of genomic variability. How can the transcriptome study improve fish management and breeding programs? Why were not more developmental stages included in this study (eleutheroembryo, yolk absorption, first-feeding larvae, pre-flexion, flexion and post-flexion larvae, and juveniles)?

Conclusion

Adjust this section according to the new discussion.

Material and methods

This section is well prepared.

How many individuals were used for each group (Celestial-Eye and Ryukin goldfish) and developmental stage? Was the transcriptome for both groups and developmental stage individually or pull processed? Please clarify these aspects.
